# The Comprehensive Detection of mRNAs, lncRNAs, circRNAs, and miRNAs in Lychee Response to Lychee Downy Blight

Mao Yin [1], Yingjie Wen [1], Juge Liu [2], Yonghua Jiang [1], Fachao Shi [1], Jiezhen Chen [1], Changhe Cai [1], Liangxi Ou [1], Qian Yan [1] and Hailun Liu [1,*]

1. Institute of Fruit Tree Research, Guangdong Academy of Agricultural Sciences, Key Laboratory of South Subtropical Fruit Biology and Genetic Resource Utilization, Ministry of Agriculture and Rural Affairs, Guangdong Provincial Key Laboratory of Tropical and Subtropical Fruit Tree Research, Guangzhou 510640, China; yanqian@gdaas.cn (Q.Y.)
2. College of Food Science and Engineering, Yangzhou University, Yangzhou 225012, China
* Correspondence: hailunl@163.com

**Abstract:** Lychee downy blight (LDB) is an oomycete pathogen (*Peronophythora litchi*) disease, which affects the fruits and leaves of lychee, resulting in economic losses. Chemical fungicides are commonly used for disease control, while for eco-safety measures, the study of molecular mechanisms involved in lychee resistance against LDB is necessary. Non-coding RNAs (ncRNAs), including long ncRNAs (lncRNAs), circular (circRNAs), and microRNAs (miRNAs), play a crucial role in plant disease resistance. To examine lychee response (fruits and leaves) to LDB, we studied the expression patterns of ncRNAs and mRNAs under LDB treatment. By whole transcriptome sequencing analyses, a total of 36,885 mRNAs, 2627 lncRNAs, 4682 circRNAs, and 525 miRNAs were identified in lychee. A differential expression (DE) analysis revealed that there were 1095 DEmRNAs, 89 DElncRNAs, 28 DEcircRNAs, and 28 DEmiRNAs in the LDB-treated fruits, as well as 1158 DEmRNAs, 132 DElncRNAs, 13 DEcircRNAs, and 197 DEmiRNAs in the LDB-treated leaves. Gene ontology (GO) and Kyoto Encyclopedia of Genes and Genomes (KEGG) analyses revealed that the potential function of DEmRNAs and the targets of ncRNAs were involved in plant defense. Furthermore, co-expression networks of putative interacting ncRNAs and mRNAs were developed, in which mRNAs encoded some receptor proteins, pointing to potential ncRNAs associated with LDB infection. Our study provided a new, brief insight to the putative role of ncRNAs in lychee response to LDB.

**Keywords:** Lychee (*Litchi chinensis* Sonn.); lychee downy blight (LDB); whole-transcriptome sequencing; mRNAs and ncRNAs; plant disease resistance; gene regulatory network





## 1. Introduction

Lychee, an important fruit in tropical and subtropical areas, is widely grown in over 20 countries across the globe [1]. It is prevalent for its smooth and juicy flesh, delicious taste, rich protein value, vitamins, folic acid, and other nutrients [2]. According to reports, the world's lychee annual production ranges from 2.6 to 2.8 million tons [3]. Given the substantial nutritional and commercial value of lychee, recently, a lot of research has been conducted on the increase of yield and postharvest shelf life [4,5]. However, as several diseases are quantitatively and qualitatively affecting the lychee industry, more measures are still needed to improve the quality of lychee.

Lychee downy blight (LDB), caused by *Peronophythora litchii* (*P. litchii*), is the primary disease affecting lychee fruits, inflorescences, and leaves, which also even extends to postharvest and transportation [6]. *P. litchii* is intrinsically necrotrophic and rampant in moist environments [7]. The infected parts (fruits and leaves) by *P. litchi* show wilting and watery brown spots, leading to severe losses in lychee production [8]. Currently, management of *P. litchii* relies on chemical fungicides, which leads to various food and environmental

safety hazards [9]. The excessive use of chemical fungicides is causing various problems in the food chain and environment [10]. Therefore, exploring the genes of lychee involved in the resistance against LDB along with their mechanisms is the basic need at the time. Previously, several methods such as microarrays, EST sequencing, and RNA sequencing (RNA-seq) have been used in LDB resistance studies [11]. For instance, researchers have used RNA sequencing technology to study the role of mRNA and miRNA in resistance to *P. litchii*. The target genes of *LcmiR159* and *LcmiR828* associated with the lychee signal transduction pathway can be altered following *P. litchii* infection, thereby activating the salicylate (SA) defense network [12]. In another study, following infection by *P. litchii*, certain genes in lychee fruits related to disease resistance or stress were found to exhibit significant upregulation [13]. However, a comprehensive analysis of the genes' regulation involved in lychee against the LDB has not been performed.

Non-coding RNAs (ncRNAs) that do not code for proteins are directly associated with the functional regulatory RNAs [14]. They are part of the RNA-protein complex involved in gene expression and can be classified into several families. These families include microRNAs (miRNAs, >200 nucleotides), long non-coding RNAs (lncRNAs, <200 nucleotides), and circular RNAs (circRNAs) [15]. The miRNAs in angiosperms not only regulate the post-transcriptional expression in its target but also participate in regulatory networks, responsible for biotic and abiotic stresses [16]. An interaction analysis of mRNAs and miRNAs in tomatoes revealed regulators of the Alternaria-stress response [17]. In the past, multiple studies have been reported where circRNAs are directly associated with gene expression on both the transcriptional and post-transcriptional levels. They act as miRNA sponges, can bind to various miRNAs, and can inhibit their interaction with target genes [18]. Additionally, gene network analyses also show the associations between circRNAs and genes involved in encoding MAP kinase and intracellular signal transduction pathways [19]. Moreover, lncRNAs have been identified as significant regulators of gene expression and play a vital role in plant defense [20]. In *Arabidopsis thaliana* (*A. thaliana*), 1832 differentially expressed lncRNAs were found after drought, cold, and abscisic acid treatment [21]. Overall, ncRNAs have broad research prospects in plant growth and stress-resistance. To date, the expression profile and function of ncRNAs during LDB infection remain largely unknown.

Whole-transcriptome sequencing analyses are beneficial to dig deeper into the transcriptional regulation in plant studies [22], and the lncRNAs, circRNAs, miRNAs, and mRNAs are usually analyzed in some plants [23]. In pepper (*Capsicum annuum*), a comprehensive bioinformatics analysis revealed that targets of pepper ncRNAs include multiple transcription factors that actively respond to cold stress by participating in bio-oxidation and phosphorylation processes [24]. Based on the expression profile, a putative interacting RNA network associated with cold was developed [24]. In poplar (*Populus*), a total of 73,012 mRNAs, 13,099 lncRNAs, 47 circRNAs, and 114 miRNAs were identified [25]. Functional analyses revealed 25 miRNA targets that were significantly enriched in the growth hormone activation signaling pathway [25]. In the beet (*Beta vulgaris*), differentially expressed ncRNAs in response to salt stress were revealed using whole transcriptome RNA-seq and degradome sequencing [26]. In addition, competing endogenous RNAs (ceRNAs) reveal the regulatory role under salt stress [26]. The whole-transcriptome sequencing analyses were widely used for plant growth, development, and stress responses.

The ncRNAs have been found to play important roles in resistance to plant diseases. However, the potential function of ncRNAs in lychee regulating LDB stress remains largely unknown. In this study, we conducted a whole-transcriptome sequencing analysis to comprehensively detect the expression patterns of ncRNAs and mRNAs in lychee infection with LDB. Our study may supply new insight into the molecular basis of how ncRNAs in lychee respond to LDB stress.



## 2. Materials and Methods

### 2.1. Plant Material and Pathogenic Bacteria

Lychee (type: Feizixiao) fruits and leaves were used as plant materials, obtained from the Institute of Fruit Trees, Guangdong Academy of Agricultural Sciences, and planted in the National Lychee Germplasm Repository, Guangzhou, China. The Feizixiao (FZX) trees used are up to 20 years old and have been fruiting steadily.

The *P. litchii* was provided by South China Agricultural University. *P. litchi* was cultured in a juice agar (CJA) medium (juice from 200 g carrot for 1 L medium, 15 g agar/L for solid media) at 27 °C. The spore suspension was filtered twice through a sterile layer and then adjusted to a concentration of $10^4$ spores/L for inoculation [11].

### 2.2. Phenotypic Identification and Assessment of P. litchii Infection

For the lychee fruit treatment, Sun's (2019) method was used with little modification [11]. In brief, the fruits were washed with sterile water three times. Each fruit was inoculated with 5 μL *P. litchii* spore suspension ($1 \times 10^4$ spores/mL). For lychee leaf treatment, Xing's (2018) method was followed with slight modifications [27]. Lychee leaves were collected and inoculated with 5 μL of *P. litchii* spore suspension ($1 \times 10^4$ spores/mL). The inoculated fruits and leaves were transferred to a fresh box and placed in an incubator at a constant temperature of $27 \pm 2$ °C. The whole experiment was conducted 3 times (3 biological replicates); each replicate consists of 10 fruits or leaves from the same lychee tree. The disease index (DI) was calculated based on a previous description [27].

### 2.3. Histochemical Staining

The reactive oxygen staining (ROS) technique was referred to in a previous description [28]. For FZX lychee leaves, they were inoculated with *P. litchii* ($4 \times 10^4$ spores/mL) or a mock (sterile water). The collected samples were then transferred to the staining solution (1 mg mL$^{-1}$ 3,3-Diaminobenzidine, Sigma, St. Louis, MO, USA) for 12 h. Afterward, the samples were decolorized in the destaining solution (70% ethanol and 30% ethanoic acid). The images were observed under a Leica M165C stereoscope.

### 2.4. Samples for Whole Transcriptome Sequencing Analysis

The FZX fruits and leaves were inoculated with 5 μL of *P. litchii* spore suspension ($1 \times 10^4$ spores/mL) or sterilized water (control, Mock). The treated leaves were collected at 24 hpi (hours post-infection). Finally, a total of 12 samples were obtained, including: six samples with leaves (*P. litchii* or Mock treatment with three replicates, respectively), and six samples with fruits (*P. litchii* or Mock treatment with three replicates, respectively).

### 2.5. RNA Library Construction, Sequencing, and Quality Control

For whole transcriptome sequencing, the total RNAs were extracted and then subjected to quality control with an Agilent 2100 Bioanalyzer (Agilent Technologies, Santa Clara, CA, USA). The libraries were constructed using the NEBNext Ultra RNA Library Prep Kit (NEB, Ipswich, MA, USA). RNA sequencing was performed by BioMarker Technologies (Beijing, China) using the Illumina NovaSeq6000 platform. Initially, Perl scripts were employed to process the raw data (in fastq format) obtained from the sequencing process. During the data-cleaning process, adapter sequences, poly-N sequences, and low-quality reads were removed from the raw data to generate clean reads. The Q20, Q30, GC content, and sequence duplication level were among the parameters used to evaluate the quality of the clean data. The high-quality clean data were mapped to the lychee reference genome, which served as the basis for all subsequent analyses [29].

### 2.6. Identification of RNAs

After identifying the coding sequences (mRNAs), the Cuffcompare software (version 2.1.1) was utilized to annotate the identified transcripts [30]. Unknown transcripts were then screened to identify putative lncRNAs. To distinguish between non-protein

coding RNA candidates and putative protein-coding RNAs, four computational tools (CNCI/CPC/Pfam/CPAT) were employed in combination. Cuffcompare software was used to select different types of lncRNAs. The find_circ tool was utilized to detect circRNAs [31]. Alignment of the sequences and mapping with mature miRNA sequences in miRbase were performed to identify known miRNAs. The prediction of novel miRNAs was carried out with the miRDeep2 tool [32].

### 2.7. Differential Expression Analysis

For the differential expression analysis, we performed the DEncRNAs and DEmR-NAs analyses using DESeq software with a screening criteria of $p$-value < 0.05 and fold change $\geq 1.5$ [33]. The gene expression patterns were graphically represented in a heatmap by the cluster analysis tool.

### 2.8. Function Annotation

For the gene function annotation, all the genes were annotated using the following databases: NCBI non-redundant (NR, ftp://ftp.ncbi.nih.gov/blast/db/FASTA/, accessed on 2 March 2023); Protein family (Pfam, http://pfam.xfam.org/, accessed on 2 March 2023); Kyoto Encyclopedia of Genes and Genomes (KEGG, http://www.genome.jp/kegg/, accessed on 3 March 2023); and Gene Ontology (GO, http://www.geneontology.org/, accessed on 3 March 2023).

The Gene Ontology (GO) enrichment analysis of the differentially expressed (DE) mR-NAs and the target genes of DElncRNAs, DEcircRNAs, and DEmiRNAs was implemented by the GOseq R packages. The KEGG analysis used KOBAS software to test the statistical enrichment of differential expression genes in KEGG pathways [34].

### 2.9. Construction of ceRNA Co-Expression Network

The ceRNA (competing endogenous RNA) co-expression network was constructed by predicting miRNA-binding RNA. Specifically, miRNAs–ncRNAs were predicted by ssearch36 (36.3.6) (https://fasta.bioch.virginia.edu/fasta_www2/fasta_down.shtml, accessed on 6 March 2023). The miRNA–mRNA pairs were predicted by TargetFinder (https://github.com/carringtonlab/TargetFinder, accessed on 6 March 2023). The networks were visualized with cytoscape software (version 3.9.1; The Cytoscape Consortium, San Diego, CA, USA).

### 2.10. Quantitative Real-Time PCR Validation

The total RNA was extracted using a Plant RNA Kit (Vazyme, Cat[#] RC401-01). Quantitative RT-PCR was performed using the SYBR Green Master Mix (Vazyme, Cat[#] Q711-02) on a LightCycler 480 Thermal Cycler (Roche, Basel, Switzerland), as described previously. *LcActin 2* was used as an internal control for sample normalization during the real-time RT-PCR analysis ($2^{-\Delta\Delta Ct}$ method). The primers used are listed in the Supplementary Data (Table S1).

### 2.11. Statistical Analyses

All statistical analyses and graph constructions were performed using GraphPad Prism 9 (GraphPad Software Inc., La Jolla, CA, USA) and R Language (R version 4.2.3, psych and ggplot2 packages). Statistical differences were performed by one-way ANOVA ($p$-value < 0.05).

## 3. Results

### 3.1. Phenotypes Analysis of Lychee Fruits and Leaves Infected by LDB

The FZX leaves and mature fruits were inoculated with *P. litchii* and their phenotypes were observed for consecutive days. Tiny mycelia on the fruits and slight browning on the leaves were observed at one dpi (day post-infection) (Figure 1A,B). Severe symptoms developed on both tissues at two dpi. Reactive oxygen species (ROS) detection confirmed

the consistent results, in which the lychee leaves drastically darkened at 1 dpi (Figure 1C). DI statistics revealed that fruits and leaves slightly changed at 1 dpi, while the fruits and leaves of FZX lychee showed a sharp increase at 2 dpi (Figure 1D,E). *LcPR1* is described as a *PATHOGENESIS-RELATED 1* (*PR1*; SA-responsive gene), which was markedly induced by LDB (Figure 1F) [28], suggesting a rapid response of FZX following LDB infection. Based on these observations, all the samples with 1 dpi (24 h) were used for the whole transcriptome sequencing analysis.

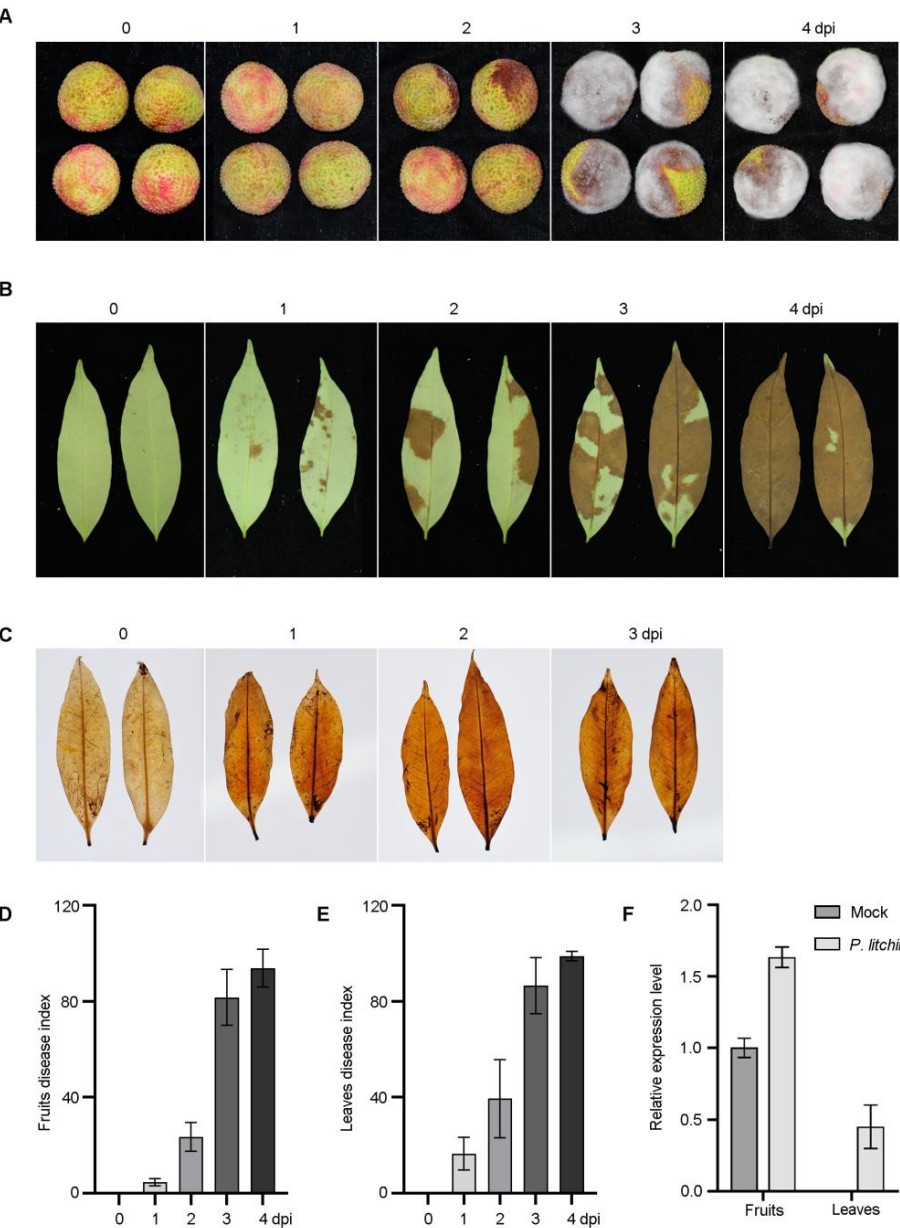

**Figure 1.** Phenotypic analysis of FZX fruits and leaves after inoculation with *P. litchii.* (**A,B**) Phenotypic investigation of fruits and leaves. (**C**) ROS detected in FZX leaves. (**D,E**) DI statistics of fruits and leaves. (**F**). Expression analysis of *LcPR1* at 1 dpi.

### 3.2. Global Response of mRNAs to P. litchii Treatment

From the RNAseq data, we identified 36,885 mRNAs in fruits and leaves based on the lychee reference genome [29]. The differential expression mRNA (DEmRNA) expression in fruits or leaves is shown in Figure 2A. The DEmRNA analysis revealed that a total of 1095 DEmRNAs (503 up-regulated, 592 down-regulated) were detected in fruits after *P. litchii* infection, while a total of 1158 DEmRNAs (669 up-regulated, 489 down-regulated)

were detected in leaves (Figure 2B, Table S2). To investigate the overlap of DEmRNAs, we utilized a Venn diagram to visualize the differential genes between fruits and leaves (Figure 2C). The results revealed that 147 DEmRNAs co-respond to LDB stress, while 948 respond in fruits and 1011 respond in leaves (Figure 2C).

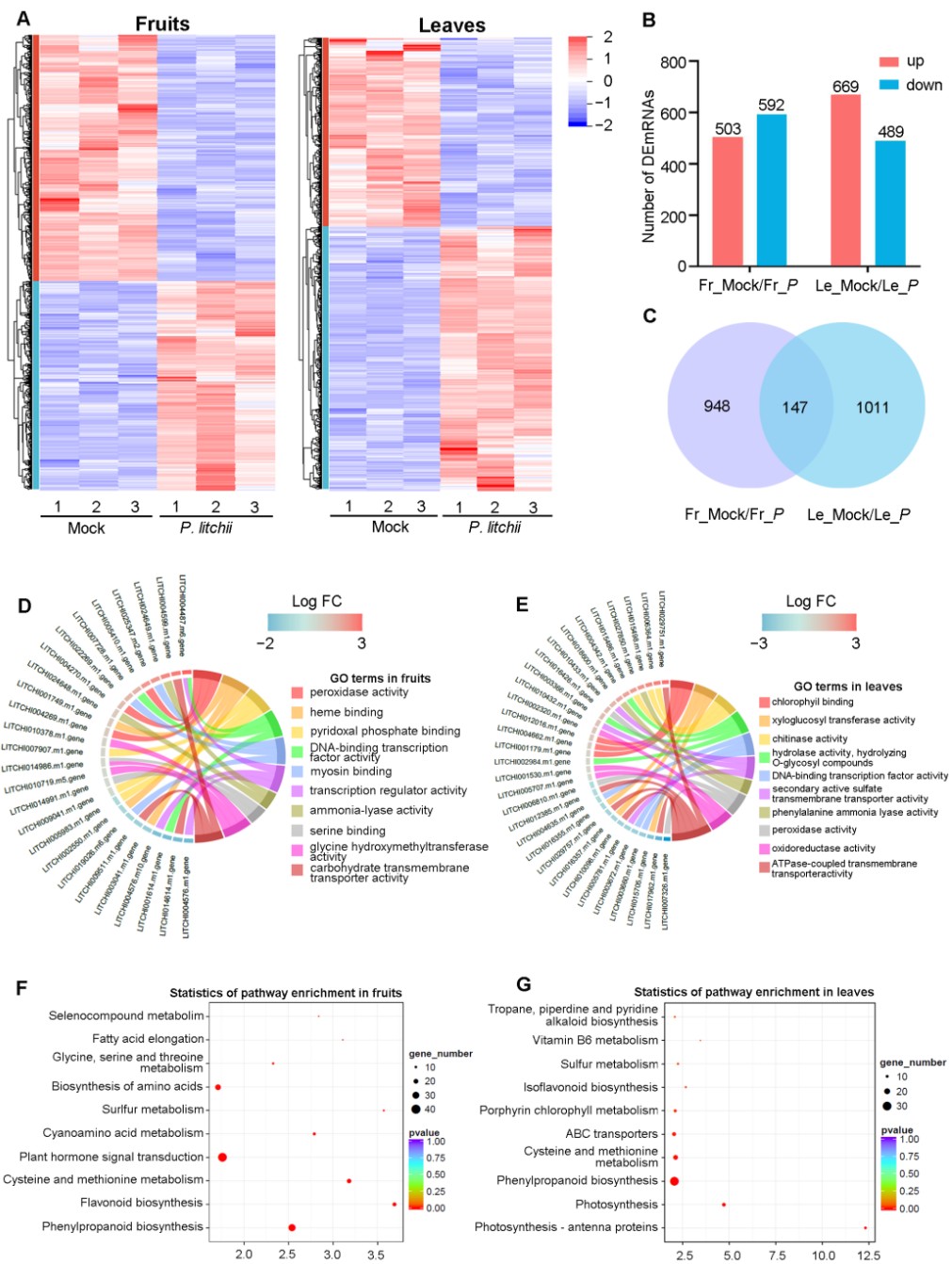

**Figure 2.** Differential expression mRNAs (DEmRNAs) analysis in lychee fruits and leaves under LDB stress. (**A**) Hierarchical clustering of DEmRNAs in fruits or leaves after *P. litchii* infection. Number 1, 2, and 3 represent three biological replicates. (**B**) The bar graphs displayed the DEmRNAs in fruits and leaves. (**C**) Venn diagram showing the overlap of DEmRNAs between fruits and leaves. (**D,E**) The network diagram generated based on GO enrichment analysis of DEmRNAs associated with molecular function. Edges in different colors represent distinct pathways. Gene nodes in different colors indicate different degrees of differential expression. (**F,G**) KEGG analysis of DEmRNAs. In (**B,C**), Fr_Mock/Fr_*P* and Le_Mock/Le_*P* represent the Mock versus *P. litchii* treatment in fruits or leaves, respectively.

The GO analysis revealed that some DEmRNAs associated with peroxidase activity, including LITCHI004269.m1, LITCHI004270.m1, LITCHI005410.m1 (Figure 2D), and LITCHI006810.m1, LITCHI007326.m1, LITCHI016426.m1 (Figure 2E). Considering that peroxidase is important in plant immune responses [35], these genes may participate in lychee resistance to LDB. In addition, the KEGG analysis revealed that a total of 116 and 117 pathways were enriched in fruits and leaves (Table S3). "Phenylpropanoid biosynthesis (ko00940)", "Cysteine and methionine metabolism (ko00270)", and "Sulfur metabolism (ko00920)" were enriched in both fruits and leaves (Figure 2F,G), which suggests these three pathways may be important for lychee resistance to LDB. Additionally, the significant pathways such as "Flavonoid biosynthesis (ko00941)", "Glycine, serine and threonine metabolism (ko00260)", and "Plant hormone signal transduction (ko04075)" were unique to fruits (Figure 2F); the "Photosynthesis-antenna proteins (ko00196)", "Photosynthesis (ko00195)", and ABC transporters (ko02010)" were unique to leaves (Figure 2G). These results indicate that these special pathways may play different functions in different tissues.

### 3.3. Global Response of lncRNAs to P. litchii Treatment

For lncRNAs, a total of 2627 candidate lncRNAs were identified from lychee fruits and leaves, including 1796 lincRNAs, 257 intronic lncRNAs, 252 antisense lncRNAs, and 322 sense_lncRNAs (Figure S1A,B). The results revealed that lncRNAs possess shorter transcripts and longer ORFs when compared to mRNAs. The lncRNAs also exhibited fewer exons compared to mRNAs (Figure S2A–F). Furthermore, the expression levels of these lncRNAs were lower than mRNAs (Figure S2G). The general patterns of DElncRNA expression in fruits or leaves are shown in Figure 3A. Upon differential expression analysis, a total of 89 DElncRNAs (35 up-regulated, 54 down-regulated) were detected in fruits under *P. litchii* infection, and a total of 132 DElncRNAs (53 up-regulated, 79 down-regulated) were detected in leaves after *P. litchii* treatment (Figure 3B, Table S2). Only three DElncRNAs (MSTRG.15653.2, MSTRG.15825.1, and MSTRG.29637.2) were identified in both tissues, while 86 and 129 DElncRNAs were identified in fruits and leaves, respectively (Figure 3C).

To investigate the biological functions of the DElncRNAs, we conducted a GO and KEGG analysis of DElncRNA target genes. The GO enrichment analysis indicated that DElncRNA targets in fruits were mainly involved in "regulation of vesicle-mediated transport", "regulation of protein-containing complex assembly", and "regulation of phagocytosis" (Figure 3D). The targets in leaves were mainly involved in "structural constituent of ribosome", "translation", and "ribosome" (Figure 3E). The KEGG analysis revealed that the targets of DElncRNAs in fruits and leaves were enriched into 106 and 133 pathways, respectively (Table S3). For fruits, the most enrichment pathways included "Non-homologous end-joining (ko03450)", "Glyoxylate and dicarboxylate metabolism (ko00630)", and "MAPK signaling pathway-plant (ko04016)" (Figure 3F). For leaves, the targets were mainly enriched to "Ribosome (ko03010)", "Oxidative phosphorylation (ko00190)", and "Pantothenate and CoA biosynthesis (ko00770)" (Figure 3G). These results suggest that these pathways may play crucial roles for lychee against LDB. However, some pathways were only detected in specific tissues, indicating that these pathways might exert specialized functions in different tissues.

### 3.4. Global Response of circRNAs to P. litchii Treatment

For the circRNA analysis, we identified a total of 4682 circRNAs from lychee fruits and leaves, and the DEmiRNA expression profiles are presented in Figure 4A. The length distributions of circRNAs are depicted in Figure S1D, with the majority of them being shorter than 3000 bp. We observed a decrease in the number of circRNAs as the sequence length increased. Previous studies reveal that there are 15 pairs of homologous chromosomes in FZX lychee (2n = 30) [30]. We found that the chromosome 13 (Chr13) contained the highest number of circRNAs (Figure S1C). Next, the DEcircRNAs analysis revealed that there were 28 DEcircRNAs (13 up-regulated and 15 down-regulated) in fruits and 13 DEcirRNAs (7 up-regulated and 6 down-regulated) in leaves (Figure 4B, Table S2). In

addition, only one DEcircRNA (Chr13:12638598 | 12653782) was co-identified in two tissues, while 27 and 12 DEcircRNAs were identified in fruits and leaves, respectively (Figure 4C). Considering that Chr13:12638598 | 12653782 responds to LDB induction in two different tissues, we speculate that it may be involved in the response of lychee resistance to LDB.

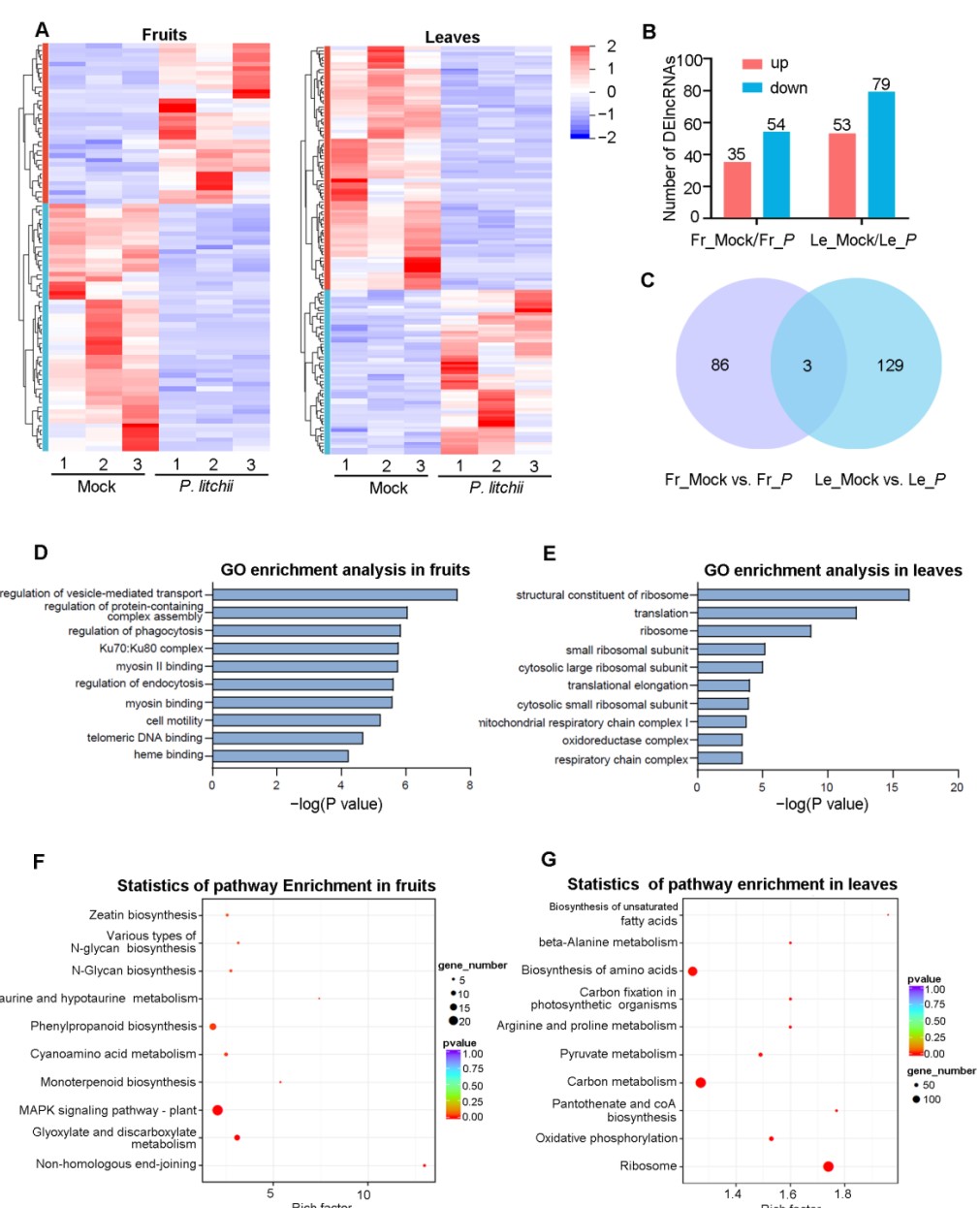

**Figure 3.** Differential expression lncRNAs (DElncRNAs) analysis in lychee fruits and leaves under LDB stress. (**A**) Hierarchical clustering of DElncRNAs in fruits or leaves following *P. litchii* infection. Number 1, 2, and 3 represent three biological replicates. (**B**) The bar graphs displayed the DElncRNAs in fruits and leaves. (**C**) Venn diagram showing the overlap of DElncRNAs between fruits and leaves. (**D,E**) GO enrichment analysis of DElncRNAs. (**F,G**) KEGG analysis of DElncRNAs. In (**B,C**), Fr_Mock/Fr_*P* and Le_Mock/Le_*P* represent the Mock versus *P. litchii* treatment in fruits or leaves, respectively.

The function of circRNAs is related to the function of the host linear transcript, and then the host gene of DEcircRNAs was further identified by the functional analysis. In particular, the GO analysis showed that the host genes of DEcircRNAs in fruits were mainly enriched in "transaminase activity", "thiamine pyrophosphate binding", "hydro-

gen peroxide catabolic process", "response to oxidative stress", and "peroxidase activity terms" (Figure 4D). However, the host genes of DEcircRNAs in leaves were mainly enriched in "phosphatidylinositol binding", "protein serine/threonine phosphatase activity", "sialyltransferase activity", "protein glycosylation", and "integral component of membrane" (Figure 4E). These results implied that these circRNAs may be involved in plants' response to stress by affecting key enzyme activity and metabolites' synthesis. The KEGG pathway analyses showed that the host genes of DEcircRNAs in fruits were enriched into four pathways, including "Glutathione metabolism, Valine (ko00480)", "leucine and isoleucine degradation (ko00280)", "Peroxisome (ko04146)", and "Phenylpropanoid biosynthesis (ko00940)" (Table S3). Meanwhile, for leaves, the host genes were only enriched in "Other types of O-glycan biosynthesis (ko00514)" (Table S3). Considering that circRNAs enriched more pathways in fruits, we speculate that they may have more intricate functions specifically within the fruits.

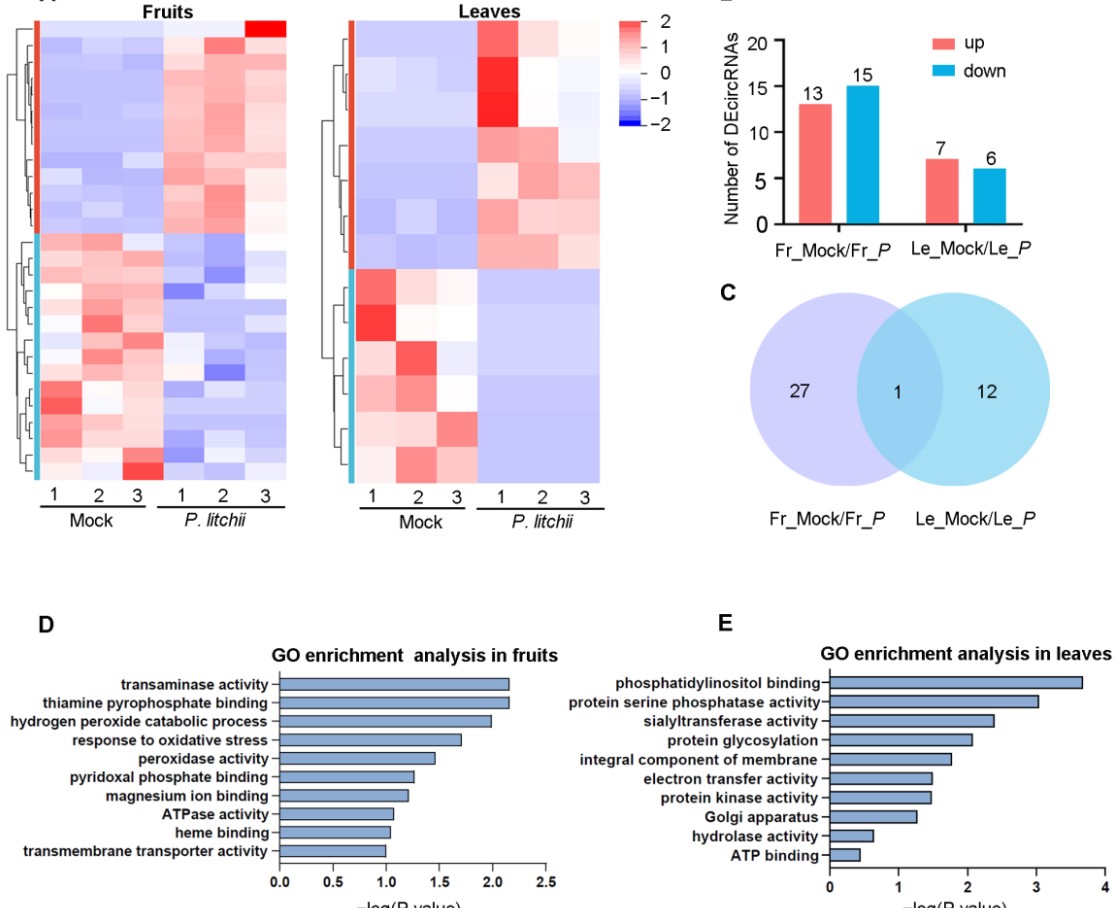

**Figure 4.** Differential expression circRNAs (DEcircRNAs) analysis in lychee fruits and leaves under LDB stress. (**A**) Hierarchical clustering of DEcircRNAs in fruits or leaves following *P. litchii* infection. Number 1, 2, and 3 represent three biological replicates. (**B**) The bar graphs displayed the DEcircRNAs in fruits and leaves. (**C**) Venn diagram showing the overlap of DEcircRNAs between fruits and leaves. (**D**,**E**) GO enrichment analysis of DEcircRNAs. In (**B**,**C**), Fr_Mock/Fr_*P* and Le_Mock/Le_*P* represent the Mock versus *P. litchii* in fruits or leaves, respectively.

### 3.5. Global Response of miRNAs to P. litchii Treatment

The analysis of sRNA-seq data revealed the identification of 525 miRNAs (consisting of 131 known miRNAs and 394 novel miRNAs), which were predominantly 20–24 nt in length (Figure S1E,F). Subsequently, the differential expression miRNA (DEmiRNA) analysis was performed (Figure 5A). More DEmiRNAs were detected after LDB treatment.

In particular, there were 28 DEmiRNAs (27 up-regulated, 1 down-regulated) that were detected in fruits, while 197 DEmiRNAs (88 up-regulated, 109 down regulated) were detected in leaves (Figure 5B, Table S2). There were 13 DEmiRNAs identified in both of the two tissues (Figure 5C), including novel_miR_238, novel_miR_172, novel_miR_230, novel_miR_332, novel_miR_92, novel_miR_160, novel_miR_35, novel_miR_303, novel_miR_49, novel_miR_107, novel_miR_208, gma-miR164a, and novel_miR_82. The co-identified miRNAs are likely to be involved in the biological processes associated with lychee against LDB.

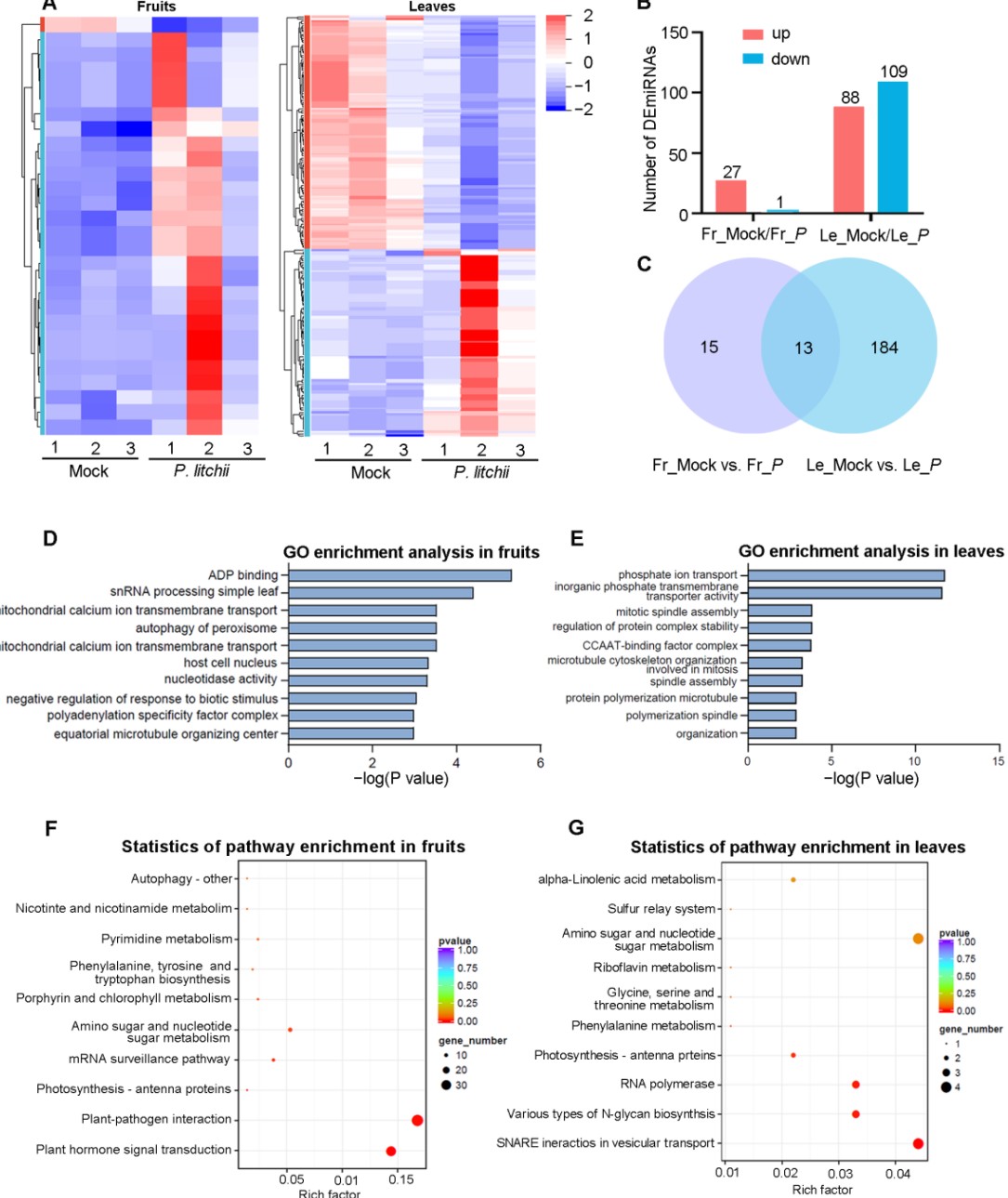

**Figure 5.** Differential expression miRNAs (DEmiRNAs) analysis in lychee fruits and leaves under LDB stress. (**A**) Hierarchical clustering of DEmiRNAs in fruits or leaves following *P. litchii* infection. Number 1, 2, and 3 represent three biological replicates. (**B**) The bar graphs displayed the DEmiRNAs in fruits and leaves. (**C**) Venn diagram showing the overlap of DEmiRNAs between fruits and leaves. (**D,E**) GO enrichment analysis of DEmiRNAs. (**F,G**) KEGG analysis of DEmiRNAs. In (**B,C**), Fr_Mock/Fr_*P* and Le_Mock/Le_*P* represent the Mock versus *P. litchii* treatment in fruits or leaves, respectively.

Next, we investigated the biological functions of DEmiRNAs in FZX fruits and leaves. The GO enrichment analysis showed that the targets of DEmiRNAs in fruits were mainly involved in "ADP binding", "snRNA processing", and "simple leaf morphogenesis" (Figure 5D). The targets in the leaves were mainly involved in "phosphate ion transport", "inorganic phosphate transmembrane transporter activity", and "mitotic spindle assembly" (Figure 5E). In addition, the KEGG analysis was performed to explore the pathways involved in DEmiRNA targets. For fruits, the targets of DEmiRNAs were enriched into 92 KEGG pathways (Table S3). The most enriched pathways were those associated with the stress response, including "Plant hormone signal transduction (ko04075)", "Plant-pathogen interaction (ko04626)", and "Photosynthesis-antenna proteins (ko00196)" (Figure 5F). For leaves, the targets were enriched into 45 KEGG pathways (Table S3). The majority targets were mainly enriched in "SNARE interactions in vesicular transport (ko04130)", "Various types of N-glycan biosynthesis (ko00513)", and "RNA polymerase (ko03020)" (Figure 5G). Next, we will conduct in-depth research on these pathways.

### 3.6. Integrative Analysis of ncRNAs and mRNA in Lychee

Based on the targeted relationship of miRNAs to obtain candidate ceRNA relationship pairs in lychee, we constructed circRNA–miRNA–mRNA and lncRNA–miRNA–mRNA co-expression networks. By considering circRNAs as a decoy, miRNAs as a center, and mRNAs as a target, the circRNAs–mRNAs network consists of 37 circRNA nodes, 47 mRNA nodes, and 109 miRNA nodes (Figure 6). Also, given that lncRNAs acts as a decoy, miRNAs as a center, and mRNAs as a target, the lncRNA–mRNA network consists of six lncRNA nodes, 65 miRNA nodes, and 44 mRNA nodes (Figure 7). We found that several miRNA156 family members occupied important nodes in both networks, such as, hbr-miR156, mdm-miR156v, and cpa-miR156a (Figures 6 and 7). It was reported that the conservative miRNA156 family is one of the most extensively researched groups of miRNAs in plants, which plays an important role in plant resistance to pathogen stress [36,37]. In addition, several mRNAs such as the encoded RING-H2 finger protein (LITCHI007260.m1), serine/threonine-protein kinase (LITCHI006900.m5), universal stress protein (LITCHI005558.m1), chaperone protein dnaJ (LITCHI014436.m1), and disease resistance protein (LITCHI013144.m1) are related to LDB stress. These findings suggest that circRNAs and lncRNAs with miRNA response-elements likely play potential regulatory roles in the mechanisms underlying lychee resistance to LDB.

### 3.7. Expression Profile Validation

A total of eight transcripts were selected for the RT-qPCR analysis to further validate their expression levels (Figure 8). The results indicated that LITCHI008759.m1 and LITCHI018040.m1 were suppressed by LDB in leaves, while LITCHI027850.m1 and Chr15:6362006 | 6368000 were rapidly induced (Figure 8A,C). Their inconsistent expression levels in fruits or leaves suggest that they may specifically respond to LDB stress in difference tissues. The MSTRG.29637.2 and MSTRG.25122.5 were down-regulated in leaves or fruits (Figure 8B,C). These findings suggest that these RNAs may have a negative regulatory effect in lychee resistance to LDB. They are potentially involved in the tissues of lychee that are associated with the anti-LDB response. The different expression patterns of these transcripts in different tissues may imply functional differentiation.

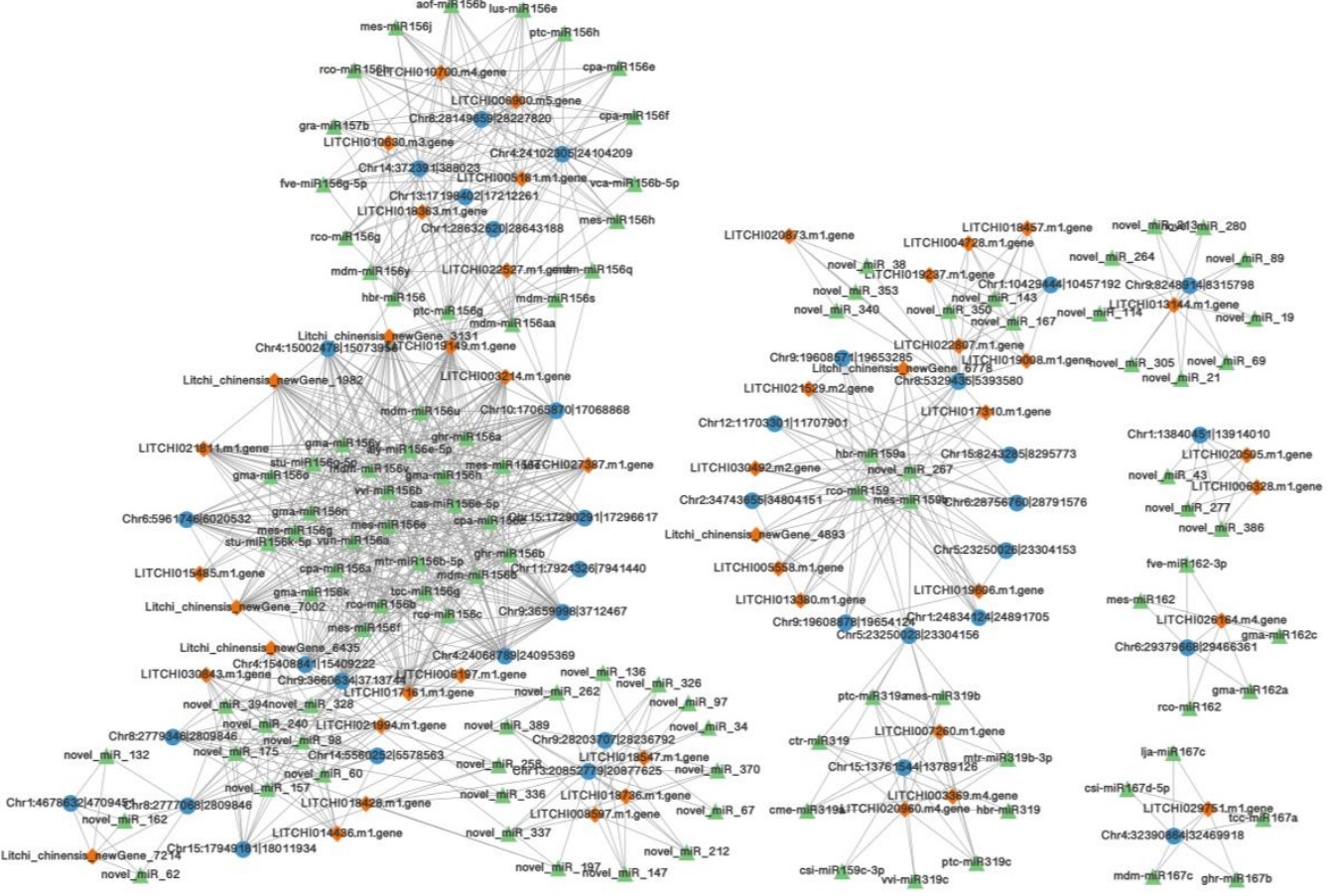

**Figure 6.** circRNA–miRNA–mRNA co-expression network in lychee. Blue rectangle, green triangle, and orange diamond represent circRNAs, miRNAs, and mRNAs, respectively.

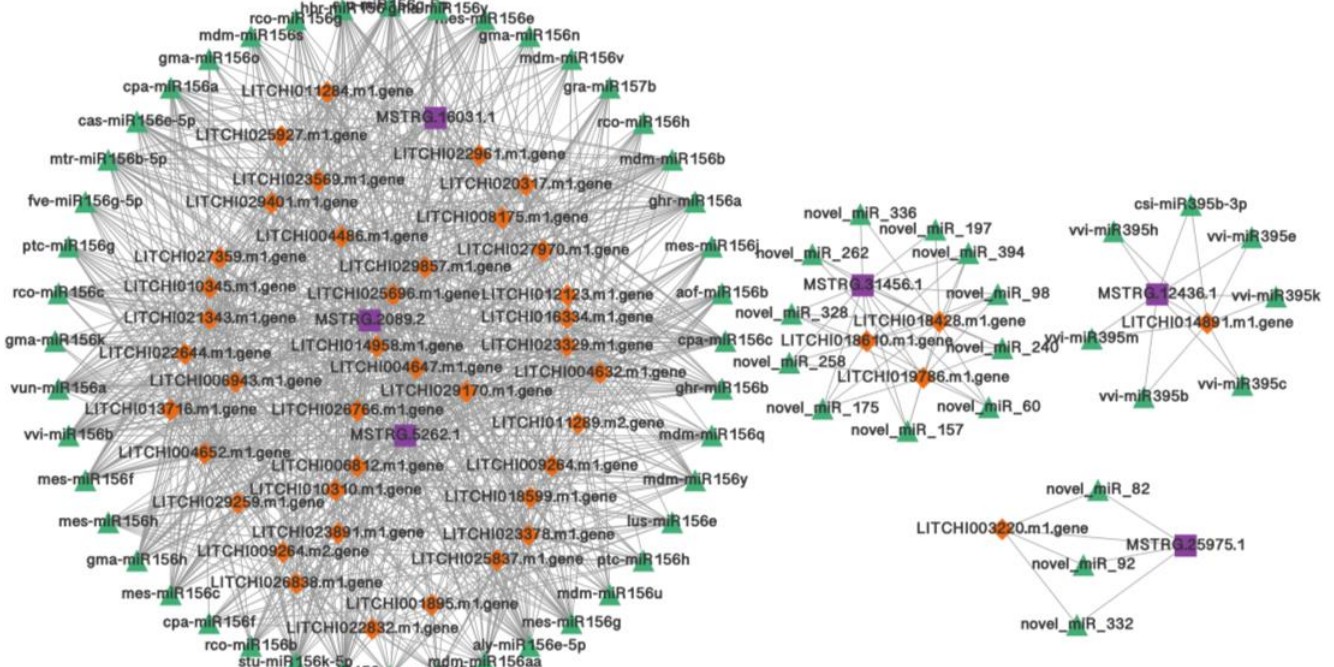

**Figure 7.** lncRNA-miRNA–mRNA co-expression network in lychee. Purple rectangle, green triangle, and orange diamond represent lncRNAs, miRNAs, and mRNAs, respectively.

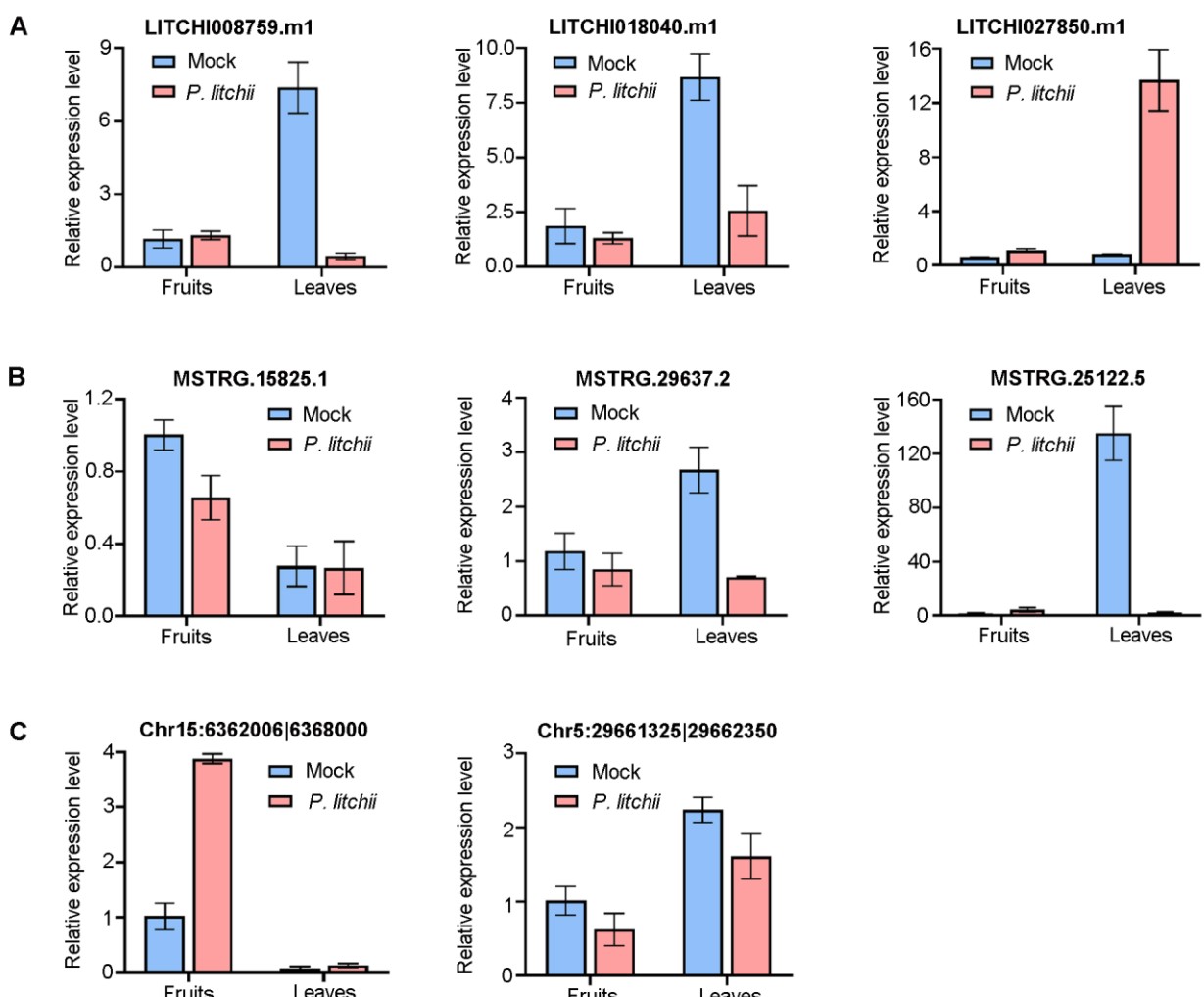

**Figure 8.** qRT-PCR analysis of ncRNAs and mRNAs. (**A**,**B**) Three RNAs each selected from mRNAs (**A**) or lncRNAs (**B**). (**C**) Two RNAs selected from circRNAs. The data are the mean ± SD of three biological replicates.

## 4. Discussion

LDB, caused by *P. litchii*, significantly impacts the quality and yield of lychee. Despite the identification of a few mRNAs involved in lychee resistance against *P. litchii* [38,39], the core molecular mechanism is still unidentified. Previous studies provide evidence of miRNA, lncRNA, and circRNA existence in different plants [40,41], but few studies have been done about the ncRNAs in lychee. In this study, we performed a whole transcriptome analysis to better understand the molecular mechanisms responsible for lychee resistance against *P. litchii* infection.

To determine the reliable treatment time, we investigated the phenotypes of FZX fruits and leaves under LDB stress. We found that both fruits and leaves rapidly responded to LDB stress during the early stages. This specifies that LDB is highly virulent, which pushes the plant to produce a strong level of stress management responses. Previous studies on lychee show strong responses for disease management, like disease resistance marker gene (e.g., *PR1*) expression, which is very high within the first 24 h of treatment [28,42]. Our results are consistent with these findings. Therefore, in current research (transcriptome analysis), the early stages' samples were collected after the treatment of LDB.

The functional analysis of DEmRNAs in lychee showed that many genes are involved in antioxidants, phenylpropanoid biosynthesis, cysteine or methionine metabolism, and sulfur metabolism processes. For example, the LITCHI004269.m1, LITCHI004270.m1,

LITCHI005410.m1, and LITCHI016426.m1 were augmented with peroxidase activities, which are quickly induced due to LDB stress. Considering antioxidants, they are important in the plant immune system [43]. These genes maybe participate in lychee resistance to LDB. We also observed different pathways in fruits and leaves. Specifically, the "flavonoid biosynthesis" and "plant hormone signal transduction" pathways were only detected in fruits, and "photosynthesis-antenna proteins", "Photosynthesis", and "ABC transporters" were only identified in leaves. "Photosynthesis-antenna proteins" and "Photosynthesis" were mainly involved in photosynthesis regulation and are closely related to plant leaf stress resistance [26,44], suggesting that the metabolic regulation of leaves primarily focuses on providing the necessary energy to support photosynthesis and helping leaves cope with LDB stress. The enrichment of specific pathways in different tissues may imply that these DEGs have additional functions in these different tissues, as plant disease resistance and growth development are dynamic processes [42].

Here, lncRNAs, mRNAs, miRNAs, and circRNAs all showed significant differences in structure, expression levels, and conservative properties in the study. lncRNAs are another type of ncRNA that is longer than 200 nucleotides and play important roles in various biological processes, including epigenetic modification, transcriptional regulation, and post-transcriptional regulation [45]. In our study, we identified a total of 2627 lncRNAs in lychee. Specifically, we found that 89 DElncRNAs in fruits and 132 DElncRNAs in leaves showed significant responsiveness to LDB stress. These findings suggest that these lncRNAs may play important roles in the lychee response to LDB stress. Previous studies reported many lncRNAs involved in plant stress. For example, 504 lncRNAs were found to be drought responsive in *Populous trichocarpa* [46], and 973 cotton DElncRNAs function in ROS scavenging [47]. Thus, lncRNAs are involved in adaptive responses to biotic stresses, but their role in the regulation of adaptive responses remains to be confirmed. Therefore, it is evident that lncRNAs play a significant role in the adaptive responses of plants to various stresses. However, their specific role in regulating stress responses, particularly in the context of biotic stress, requires further confirmation and investigation.

circRNAs are stable non-coding RNAs with closed circular structures that contain microRNA binding sites, allowing them to sequester cytoplasmic miRNAs and relieve their inhibitory effects on gene expression [48]. Similarly, we identified a total of 4682 circRNAs in FZX, in which 28 or 13 DEcircRNAs were identified in lychee fruits or leaves. These findings provide valuable insights into the role of circRNAs in lychee and their potential contribution to fruits and leaves in LDB stress. circRNAs exhibit various biological functions in plants. Previous studies have reported 62 circRNAs related to plant growth and survival in wheat [49]. In tomatoes, 163 circRNAs were associated with low-temperature stress [50]. Currently, there is limited research on circRNAs in lychee. However, we have identified various circRNAs that were involved in the lychee response to LDB. These newly identified circRNAs not only provide potential references for future studies but also pave the way for further research in lychee resistance to LDB.

miRNAs are also ncRNAs which regulate post-transcriptional events and take part in various biological processes [51]. Here, we conducted miRNA profiling in lychee and detected a total of 525 miRNAs. Among them, 28 miRNAs were specifically identified in fruits, while 197 miRNAs were specifically identified in leaves. The difference in the number of DEmiRNAs between lychee fruits and leaves may be attributed to the distinct functions they perform in different tissues. Lychee leaves may have a higher susceptibility to pathogen stress or be more inclined to generate disease-resistance responses. Previous studies have primarily focused on the role of miRNAs in regulating plant growth and development [52], with limited attention given to plant disease resistance. Our research expands the scope of miRNA studies by focusing on their involvement in plant defense mechanisms. Furthermore, our findings provide valuable references for understanding the role of miRNAs in lychee resistance to LDB.

To explore the potential defense function of DEncRNAs in this study, the GO and KEGG analyses of DEncRNA-targets were performed in this study. We noticed that DEncR-

NAs are related to the defense response, especially the regulation of biological stimulus-response. For example, some DElncRNA targets (LITCHI004773.m1 and LITCHI012609.m1) in fruits annotated to threonine-protein kinase were enriched in "MAPK signaling pathway-plant" after LDB infection. Taking into account the important role of the "MAPK signaling pathway-plant" in plant disease resistance [53], these DElncRNAs (MSTRG.5897.86, MSTRG.5897.86, and MSTRG.5897.86) in fruits may play a key role in lychee fruits' resistance against LDB by regulating their targets. Interestingly, this pathway was not observed in the leaves, implying a potential differentiation in disease resistance functionality among different tissues of the lychee plant. The host genes of DEcircRNAs in leaves were mainly enriched in "phosphatidylinositol binding", "protein serine/threonine phosphatase activity", "sialyltransferase activity", "protein glycosylation", and "integral component of membrane". These results implied that these circRNAs may be involved in plant responses to LDB stress by affecting key enzyme activity and metabolites synthesis. Additionally, we observed that some pathways were not co-enriched in leaves and fruits. These specific pathways enriched in diverse tissues may play distinct roles in lychee resistance against LDB.

Moreover, we constructed circRNA–miRNA–mRNA and lncRNA–miRNA–mRNA co-expression networks, and some candidate mRNAs and ncRNAs associated with *P. litchii* stimulation were explored. The networks showed that several mRNAs encoded receptor proteins, including zinc finger proteins, disease-resistant proteins, signal transduction-related proteins, etc. These genes (LITCHI007260.m1, LITCHI006900.m5, LITCHI005558.m1, and LITCHI014436.m1) may play vital roles in plant and pathogen interactions, which could be related to lychee resistance to *P. litchii*. Furthermore, present research reported that miRNA156 family members play a crucial role in plant response to pathogens [36,37,54]. miR-156 is crucial in pine trees infected with stem rust, and their target genes are significantly suppressed [55]. Here, most of the miRNA156 family members were found to be important nodes in two networks. Taken together, these findings suggest a potential role of ncRNAs in regulating host–pathogen interactions. However, further experiments are required to confirm the network's interactions between ncRNAs and mRNAs.

## 5. Conclusions

LDB poses a formidable challenge to lychee production and postharvest storage. It was necessary to conduct a whole transcriptome analysis to study lychee resistance to LDB. In the current study, 36,885 mRNAs, 2627 lncRNAs, 4682 circRNAs, and 525 miRNAs were identified in lychee by whole transcriptome sequencing analysis. Among them, 1095 DEmRNAs, 89 DElncRNAs, 28 DEcircRNAs, and 28 DEmiRNAs were detected in fruits, and 1158 DEmRNAs, 132 DElncRNAs, 13 DEcircRNAs, and 197 DEmiRNAs were detected in leaves. The GO and KEGG analyses revealed that some DEmRNAs and ncRNAs were involved in plant defense. The construct of the ceRNA co-expression networks implied a putative interaction between ncRNAs and mRNAs in lychee associated with LDB infection. Gene expression revealed that three mRNAs (TCHI008759.m1, LITCHI018040.m1, and LITCHI027850), two lncRNAs (MSTRG.15825.1 and MSTRG.29637.2), and one circRNA (Chr15:6362006|6368000) respond to LDB stress in FZX fruits or leaves, suggesting that these RNAs may be important for lychee resistance to LDB.

**Supplementary Materials:** The following supporting information can be downloaded at: https://www.mdpi.com/article/10.3390/agronomy13071904/s1, Figure S1. The characterization of ncRNAs. (**A**) Screening results of lncRNAs by cpc, cnci, pfam and cpat. (**B**) The classification of lncRNAs. (**C**) Chromosome distribution of circRNAs. (**D**) Length distribution of circRNAs. (**E**) Length distribution of known miRNAs. (**F**) Length distribution of novel miRNAs; Figure S2. Comparison of structures and expression levels between lncRNAs and mRNAs. (**A**) and (**B**) Density distribution of transcript lengths in lncRNAs and mRNAs. (**C**) and (**D**) Density distribution of the number of ORFs in lncRNAs and mRNAs. (**E**) and (**F**) Density distribution of the number of exons in lncRNAs and mRNAs. (**G**) Comparison of expression levels between lncRNAs and mRNAs; Table S1. Primer

of RT_qPCR; Table S2. Summary of differential expression mRNAs and ncRNAs; Table S3. KEGG analysis of differential expression mRNAs and ncRNAs.

**Author Contributions:** H.L. conceived and designed the research. M.Y. and Q.Y. performed the data analysis and drafted the manuscript. M.Y., Y.W., J.L., Y.J. and F.S. helped in the analysis of the data. J.C., C.C. and L.O. reviewed the manuscript. All authors have read and agreed to the published version of the manuscript.

**Funding:** This work was funded by the Key-Area Research and Development Program of Guangdong Province (2022B0202070002), the Natural Science Foundation of Guangdong Province (2021A1515011031), and the National Natural Science Foundation of China (32102333).

**Data Availability Statement:** The data sets supporting the results of this article are deposited in the National Center for Biotechnology Information (NCBI) repository under GEO (Gene Expression Omnibus) accession: GSE222652. Genomic sequences and gene annotation information of lychee were downloaded from http://www.sapindaceae.com/, accessed on 26 February 2023.

**Conflicts of Interest:** The authors declare that they have no known competing financial interests or personal relationships that could have appeared to influence the work reported in this paper.

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
