# Peer review of "The Comprehensive Detection of mRNAs, lncRNAs, circRNAs, and miRNAs in Lychee Response to Lychee Downy Blight"

_agronomy, doi:10.3390/agronomy13071904_

Round 1

Reviewer 1 Report

The Authors have done extensively nice piece of work. They have performed whole transcriptome analysis of P. litchi infected and non-infected leaves and fruits of Lychee and identified 36,885 mRNAs, 2,627 lncRNAs, 4,682 circRNAs, and 525 miRNAs. The expression profile analysis revealed that after infection, there were 1,095 DEmRNAs, 89 DElncRNAs, 28 DEcircRNAs, and 28 DEmiRNAs in the fruits, as well as 1,158 DEmRNAs, 132 DElncRNAs, 13 DEcircRNAs, and 197 DEmiRNAs in the leaves. They have also performed few bioinformatics analysis and reported their significant pathways and roles in the organism. Such analysis gives strong background about the up-regulation and down-regulation of genes during infection. I recommend this manuscript for the publication in MDPI Agronomy journal after revision but I have some concern regarding work and the manuscript which is as follows:

 Major Comments:

1.     In Materials and Method Section: Line no. 83-86: After planting the Lychee how long it has taken to grow fully and produce fruits? How old the Lychee tree/plant was at the time of infection? Plant age is important to know for other researchers in the field.

2.     Line no. 92-98: Whether fruits and leaves were separated from the tree or infection was done in the live tree?

3.     If infection was done separately, why it was NOT done in live condition i.e. within tree leaves and fruits itself to know the up-regulation and down-regulations of genes in live condition as it is done in plant species?

4.     Line no. 116-118: generally for transcriptome analysis samples are collected at different time points after infection such as 6-12hrs post infection, 24 hrs post infection and 36-48 hrs post infection. Why only one time point i.e. 24 h after infection is used? As per many reports, 12 hrs post infection is very important for transcriptome analysis which is missing in this study.

5.     Usually, after 1-2 weeks of infection different types of severe symptoms appeared on the different plant/tree species but authors have not taken any phenotypic observation after 1-2 weeks of infected leaves and fruits? This aspect is very important to understand the disease and pathogen severity.

6.     Line no. 174-176: ………. We found that a total 76 DEmRNAs common response to P. litchii stress, and there were 948 and 1011 DEmRNAs uniquely identified in fruits, and leaves respectively (Figure 2C) but in Figure 2C it shown that 147 common DEmRNAs. This needs to be corrected.

7.     Figure 8 have a total of 8 graphs for expression profile but it is not elaborated significantly. What message authors have got from this analysis? This must be explained as this is one of the main data from the work.

Minor Comments:

1.     The full name of pathogen genus P. litchii must be given in the beginning of manuscript, when it is used first time.

2.     Line no. 67: full name of Arabidopsis thaliana must be given.

3.     Line no. 104-105: Sentence structure needs to be modified as it looks incomplete sentence.

4.     Three replicates were used and each replicates had 10 leaves. Whether these 3 replicates were from same tree or different tree. Needs to be clearly mentioned.

5.     Line no. 112: the word “identify” may be replaced with “identification”.

6.     Line no. 116: how a total of 12 samples ……….. not clear.

7.     Line no. 132 and 136: two softwares are written as “gffcompare” and “cuffcompare”. Whether both are same or it is typo error?

8.     In Figure 2-5 A: the number 1, 2 and 3 is mentioned in Heatmap. What does it mean? Is it replicate no. 1, replicate no. 2 and replicate no. 3?? Is shd be clear.

9.     Line no. 199: the word “significantly” may be replaced with “significant”.

10.                        Line no. 201: Figure 2H ………. This Figure is NOT existing in figure section.

11.                        The manuscript should be re-checked for any English grammar mistakes and should be corrected.

12.                        Figure no. 3 D, E, F, G: fruit shd be replaced with “fruits” and leaf shd be replaced with “leaves”.

13.                         Line no. 233-244: authors have reported the global response of pathogen on different chromosomes. It will be good if authors also mentioned about the total number of chromosomes present on Lychee genome.

14.                        Line no. 265: If authors are writing “novel miRNAs” what is the basis for indicating “novel” it may be “new” in stead of “novel” or else plz justify the nonelity.

15.                        Line no. 270: Figure 4B……. this figure belong to DEcircRNA and NOT miRNA. Plz check.

16.                        Figure 5B is NOT mentioned anywhere in the manuscript.

17.                        Line no. 291: Figure 5H is NOT available in figure section.

18.                        Figure 5F is NOT mentioned anywhere in the manuscript.

19.                         What is the significance and important message comes from the figure 6 network. It should be elaborated.

The manuscript should be re-checked for any English grammar mistakes and should be corrected. Some places fruit shd be replaced with “fruits” and leaf shd be replaced with “leaves”.

Some correction should be made for present tense/Past tense. The use of words "is" "are" "was" were" should be properly checked.

Reviewer 2 Report

The manuscript by Yin et al has profiled various RNAs in lychee in response to lychee downy blight. Due to the advances in sequencing techniques, these days sequencing has become a common thing and I see a lot of reports on "omics". I see one common problem in the majority of reports using high throughput technology. The interpretation and proper discussion of results is missing. The report mentioned expression of different pathways in leaves, fruits etc but the significance of the pathways is not sufficiently discussed. For example, in page 7 line 196-198, if you say certain pathways were unique to fruits, also the authors should justify why it could be unique. It is just a representative example. I noticed the authors mentioning the construction of circRNA-miRNA-mRNA and lncRNA-miRNA-mRNA coexpression networks. What is the outcome of the network? Again how significant the data is in the context of biology.    Apart from that I see the initial full form of abbreviations missing for P. litchii and ceRNA. In the Figures, the meaning of Fr_Mock/Fr_P and Le_Mock/Le_P is missing (Apologies if I fail to notice.    The authors are sincerely requested to address the aforesaid comments. 

Reviewer 3 Report

The manuscript addresses an interesting topic. However, it is necessary to include some changes in order to improve its structure and presentation.

Abstract

1-      Rebuilt the abstract with a clear introduction (clear problematic), which is suggested to show the importance of your study.

2-      Reformulate the objective and methodology with clear and brief sentences.

3-      In summary, mention the projection of your results for agronomic purposes.

4-      Be careful with language issues.

Introduction

1.       Line 28 please add some statistics about annual global production of lychee fruit.

2.       Line 30: add references.

3.       Line 44: add references.

4.       Line 71 To date, the expression profile and function of ncRNA during P. litchii infection remain largely unknown. Please add studies that used the same method on other plants and cite their advantages.

5.       Be careful with language ( I noted many issues in the text).

Methods

1.       Please add section about gene functional annotation and co-expression network.

2.       Please add a separate statistical analysis section with used tests and their details.

Results

Please detail your results with a concentration on the most relevant findings to show the value of your study.

Discussion

The discussion is short and poor. Please put your relevant findings and discuss them with references that have similar results on the other species or other aspects. Use more recent references.

Extensive editing of English language required
